# Divergent rhodium-catalyzed electrochemical vinylic C–H annulation of acrylamides with alkynes

Yi-Kang Xing[1,4], Xin-Ran Chen[2,4], Qi-Liang Yang[3,4], Shuo-Qing Zhang [2], Hai-Ming Guo[3], Xin Hong [2✉] & Tian-Sheng Mei [1✉]

α-Pyridones and α-pyrones are ubiquitous structural motifs found in natural products and biologically active small molecules. Here, we report an Rh-catalyzed electrochemical vinylic C–H annulation of acrylamides with alkynes, affording cyclic products in good to excellent yield. Divergent syntheses of α-pyridones and cyclic imidates are accomplished by employing *N*-phenyl acrylamides and *N*-tosyl acrylamides as substrates, respectively. Additionally, excellent regioselectivities are achieved when using unsymmetrical alkynes. This electrochemical process is environmentally benign compared to traditional transition metal-catalyzed C–H annulations because it avoids the use of stoichiometric metal oxidants. DFT calculations elucidated the reaction mechanism and origins of substituent-controlled chemoselectivity. The sequential C–H activation and alkyne insertion under rhodium catalysis leads to the seven-membered ring vinyl-rhodium intermediate. This intermediate undergoes either the classic neutral concerted reductive elimination to produce α-pyridones, or the ionic stepwise pathway to produce cyclic imidates.

---

[1] State Key Laboratory of Organometallic Chemistry, Center for Excellence in Molecular Synthesis, Shanghai Institute of Organic Chemistry, University of Chinese Academy of Sciences, Chinese Academy of Sciences, Shanghai, China. [2] Department of Chemistry, Zhejiang University, Hangzhou, China. [3] Henan Key Laboratory of Organic Functional Molecules and Drug Innovation, Henan Normal University, Xinxiang, Henan, China. [4] These authors contributed equally: Yi-Kang Xing, Xin-Ran Chen, Qi-Liang Yang. ✉email: hxchem@zju.edu.cn; mei7900@sioc.ac.cn

α-Pyridones and α-pyrones are ubiquitous structural motifs found in natural products and biologically active small molecules[1–3]. Transition metal-catalyzed vinylic C–H annulation of acrylic amides or acrylic acids with alkynes has recently emerged as one of the most powerful tools for their synthesis[4–15]. In 2009, Miura and co-workers described an early example of Rh-catalyzed oxidative coupling of substituted acrylic acids with alkynes using $Ag_2CO_3$ as the oxidant, affording α-pyrones[16]. Subsequently, the groups of Li and Rovis reported Rh-catalyzed vinylic C–H annulation of acrylamides with alkynes to afford α-pyridones under elevated temperature using stoichiometric transition metal oxidants (Fig. 1a, left side)[17,18]. Inspired by these seminal works, various transition metal-catalyzed vinylic C–H annulation reactions with alkynes have been developed to prepare α-pyridones or α-pyrones, including ones catalyzed by Rh[19–24], Ru[25–29], Co[30–33], Pd[34–36], and Fe[37,38] catalysts. Despite these advances, important challenges remain, including: (1) typically high reaction temperatures (100–120 °C); (2) stoichiometric transition metal oxidants such as $Cu(OAc)_2$ or AgOAc are generally required to regenerate catalysts; (3) a highly selective divergent synthesis of α-pyridones and cyclic imidates (Fig. 1a, right side) from acrylamides is still lacking[17].

**Fig. 1 Rh-catalyzed vinylic C–H annulation of acrylamides with alkynes to afford α-pyridone and cyclic imidate. a** Rh-catalyzed vinylic C–H annulation under elevated temperature with stoichiometric transition metal oxidants. **b** Rh-catalyzed electrochemical vinylic C–H annulation of acrylamides with alkynes.

**Table 1 Annulation optimization with acrylamide 1a and diphenylacetylene[a].**

| Entry | 1(R) | Variation from standard conditions | Yield[b](%) of 3 | Yield[b](%) of 4 |
|---|---|---|---|---|
| 1 | **1a** (Ts) | None | 99(91)[c] | n.d. |
| 2 | **1a** | CH₃CN instead of MeOH | 98 | n.d. |
| 3 | **1a** | DMF instead of MeOH | 20 | n.d. |
| 4 | **1a** | CF₃CH₂OH instead of MeOH | 17 | n.d. |
| 5 | **1a** | HFIP instead of MeOH | 9 | n.d. |
| 6 | **1a** | NaOAc instead of n-Bu₄NOAc | 86 | n.d. |
| 7 | **1a** | NaOPiv instead of n-Bu₄NOAc | 82 | n.d. |
| 8 | **1a** | No (Cp*RhCl₂)₂ | n.d. | n.d. |
| 9 | **1a** | No electric current | <5 | n.d. |
| 10 | **1a** | IKA ElectraSyn 2.0 | 99[d](92)[c] | n.d. |
| 11 | **1a** | Graphite(+) II Pt(−) | 91[d] | n.d. |
| 12 | **1a** | Graphite(+) II Graphite(−) | 95[d] | n.d. |
| 13 | **1b** (Ph) | None | <5 | **4b** (95)[c] |
| 14 | **1c** (p-NO₂-C₆H₄) | None | <5 | **4c** (86)[c] |
| 15 | **1d** (p-OMe-C₆H₄) | None | <5 | **4d** (42)[c] |

[a]Reaction conditions: **1a** (0.3 mmol), **2a** (0.2 mmol), (Cp*RhCl₂)₂ (4 mol%), n-Bu₄NOAc (3.0 equiv.) and MeOH (3 mL), in an undivided cell with two platinum electrodes (each 1.0 × 1.0 cm²), 60 °C, 1.5 mA, 7 h.
[b]The yield was determined by ¹H NMR using 1,4-dimethoxybenzene as an internal standard.
[c]Isolated yield.
[d]**1a** (0.3 mmol), **2a** (0.2 mmol), (Cp*RhCl₂)₂ (4 mol%), n-Bu₄NOAc (3.0 equiv.), and MeOH (6 mL) in an undivided cell with two electrodes (each 3.0 × 0.8 cm²), room temperature, 1.5 mA, 7 h, n.d. not detected.

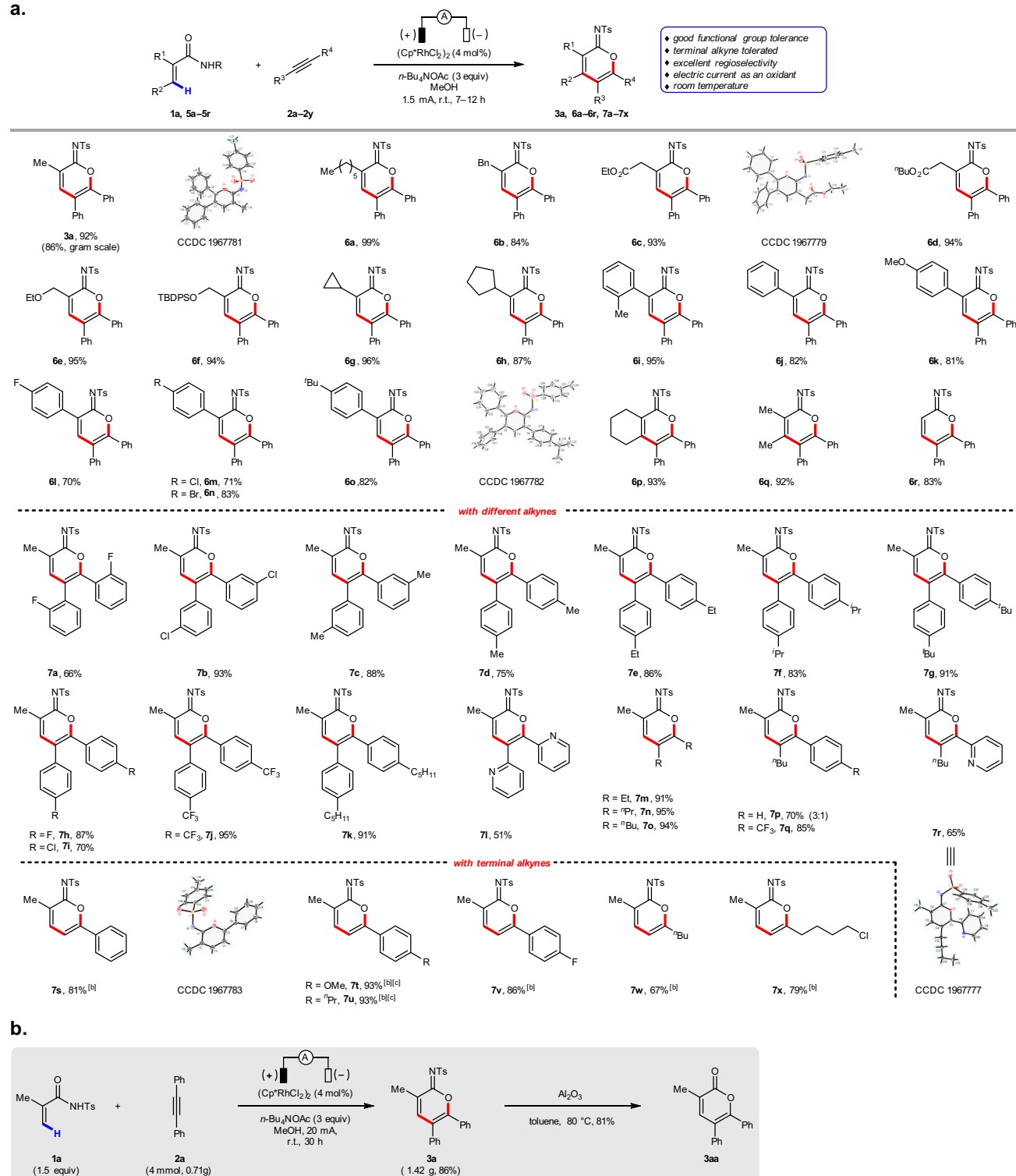

**Fig. 2 Scope of cyclic imidates and synthetic application. a** Substrate scope synthesis of cyclic imidates with IKA Electrasyn 2.0. [a]Isolated yields are reported. Reaction conditions: **1a** or **5** (0.3 mmol), **2** (0.2 mmol), (Cp*RhCl₂)₂ (4 mol%), n-Bu₄NOAc (3.0 equiv.), and MeOH (6 mL) in an undivided cell with two platinum electrodes (each 3.0 × 0.8 cm²), room temperature, 1.5 mA, 7–12 h. [b]**5** (0.15 mmol), **2** (0.3 mmol). [c]The reaction was carried out at 60 °C. **b** Gram-scale experiment and synthetic application.

Electrochemical organic synthesis has received tremendous attention because electric current offers an environmentally benign alternative to conventional methods for oxidation and reduction of organic compounds, such as those involving chemical oxidants and reductants[39–60]. Transition metal-catalyzed

electrochemical arene C–H annulation with alkynes has been developed using catalysts including Co[61–64], Ru[65–71], Rh[72–74], and Cu[75]. In contrast, electrochemical vinylic C–H annulation with alkynes is less studied. Recently, we reported an Ir-catalyzed electrochemical vinylic C–H annulation reaction of acrylic acids

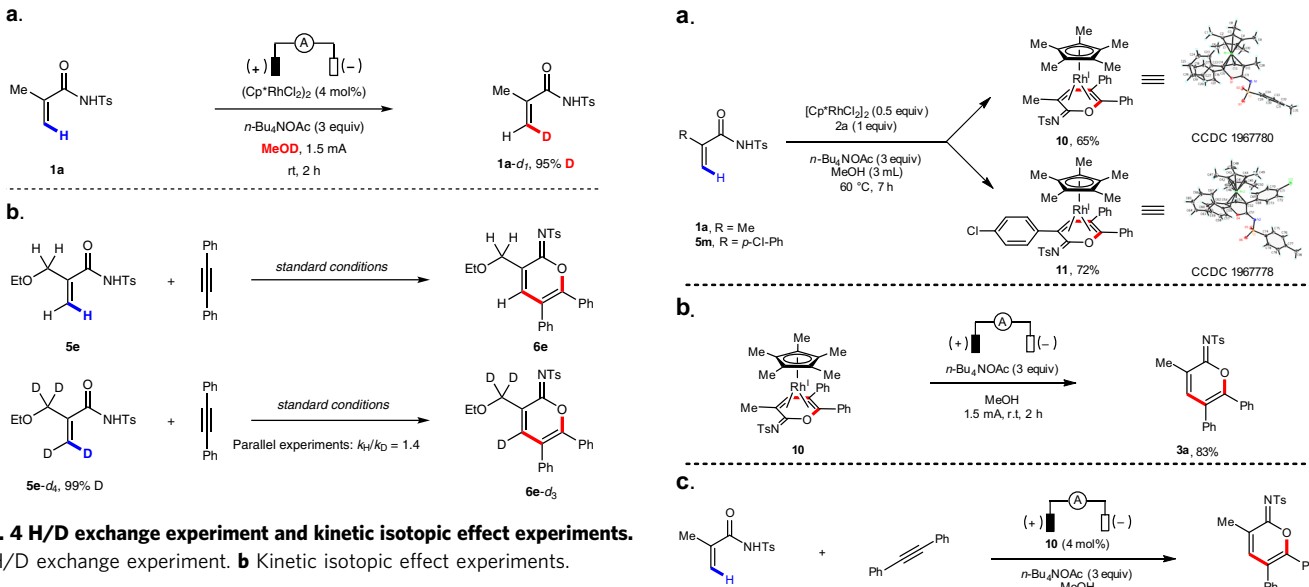

**Fig. 3 Scope of α-pyridones.** Isolated yields are reported. Reaction conditions: **1b** or **8** (0.3 mmol), **2** (0.2 mmol), (Cp*RhCl₂)₂ (4 mol%), n-Bu₄NOAc (3.0 equiv.) and MeOH (3 mL), in an undivided cell with two platinum electrodes (each 1.0 × 1.0 cm²), 60 °C, 1.5 mA, 7–12 h.

**Fig. 4 H/D exchange experiment and kinetic isotopic effect experiments.**
**a** H/D exchange experiment. **b** Kinetic isotopic effect experiments.

**Fig. 5 Stoichiometric reaction. a** Stoichiometric reactions in the absence of electric current. **b** Anodic oxidation of complex **10**. **c** Catalytic reaction with complex **10**.

with internal alkynes, affording α-pyrones in good yields, but terminal alkynes are not tolerated[76]. Subsequently, Ackermann and co-workers demonstrated Ru-catalyzed electrochemical vinylic C–H annulation of acrylamides with symmetric internal alkynes at elevated temperature (140 °C)[77]. Herein, we report an Rh(III)-catalyzed electrochemical vinylic C–H annulation of acrylamides with alkynes in an undivided cell under mild reaction conditions. Importantly, divergent syntheses of α-pyridones and cyclic imidates are achieved by varying the N-substituent of the acrylamides. Furthermore, terminal alkynes are well tolerated in this Rh-catalyzed electrochemical vinylic C–H annulation (Fig. 1b). We also probed the reaction mechanism by carrying out cyclic voltammetric analysis and kinetic isotopic experiments. Density functional theory (DFT) calculations elucidated origins of substituent-controlled chemoselectivity. The sequential C–H activation and alkyne insertion under rhodium catalysis leads to the seven-membered ring vinyl-rhodium intermediate. This intermediate undergoes either the classic neutral concerted reductive elimination to produce pyridones, or the ionic stepwise pathway to produce cyclic imidates. The electronic nature of the N-substituent has exactly the reversal effect on the rates of neutral

concerted and ionic stepwise reductive elimination pathways, which switches the chemoselectivity.

## Results

**Optimization studies.** Initially, we probed various reaction conditions using 2-methylacrylamide (**1a**) and diphenylacetylene (**2a**) as reaction partners in an undivided cell (Table 1 and Supplementary Tables 1–7). To our delight, using (Cp*RhCl₂)₂ as the precatalyst, n-Bu₄NOAc as the electrolyte, and MeOH as the solvent in an undivided cell with two platinum electrodes under constant-current electrolysis at 1.5 mA for seven hours at 60 °C, cyclic imidate **3a** can be obtained in 91% isolated yield (Table 1, entry 1). Acetonitrile as solvent affords a similar yield, while yield diminishes significantly when other solvents are used (entries 2−5). Other electrolytes such as NaOAc and NaOPiv result in slightly lower yields (entries 6 and 7). Control experiments show that no significant amount of annulation product is produced in

the absence of the catalyst or electric current (entries 8 and 9). To our delight, 92% isolated yield is obtained when the reaction is carried out with IKA ElectraSyn 2.0 at room temperature (entry 10)[78]. Furthermore, changing the electrode material caused a small decrease in yield (entries 11 and 12). Interestingly, switching to the synthesis of α-pyridones instead of cyclic

imidates can be achieved by simply changing the N-substitution of acrylamides (entries 13–15). α-Pyridone **4b** can be obtained in 95% isolated yield when N-phenyl acrylamide **1b** is used (entry 13). Other N-aryl groups afford lower yields with good selectivity of α-pyridones versus cyclic imidates (entries 14 and 15).

### Scope of cyclic imidates

**Scope of cyclic imidates**. With the optimized reaction conditions in hand, we investigated the generality of this electrochemical C–H annulation. As shown in Fig. 2a, various acrylamides substituted with alkyl, ester, ether, aryl, fluoro, chloro, and bromo groups are well tolerated, affording the corresponding cyclic imidates in good to excellent yields (**3a**, **6a–6r**). Unfortunately, β-substituted substrate like cinnamide-derived acrylamides give lower yields, which could be due to the steric effects (see Supplementary information for more details). A variety of alkynes react well, including diarylacetylenes (**7a–7l**) and dialkylacetylenes (**7m–7o**). With unsymmetrical alkynes, regioselectivity is governed by arene electronics. For example, moderate regioselectivity is achieved with n-butyl phenyl acetylene (**7p**).

In contrast, excellent regioselectivities are obtained when electron-deficient arylacetylenes are employed (**7q** and **7r**). In addition, excellent regioselectivities and yields are accomplished using terminal alkynes, with the alkyl or aryl groups oriented proximal to the oxygen heteroatom in the product (**7s–7x**). (As a reminder, terminal alkynes are not tolerated in the aforementioned Ir-catalyzed electrochemical C–H annulation[76].) Furthermore, the structures of **3a**, **6c**, **6o**, **7r**, and **7s** were unambiguously verified by X-ray analysis. Finally, we demonstrated the preparative utility of this Rh-catalyzed electrochemical C–H annulation reaction by running a reaction containing 6.0 mmol of substrate **1a** and 4.0 mmol of substrate **2a** to afford cyclic imidate **3a** in 86% yield, which can be further converted into α-pyrone **3aa** (Fig. 2b).

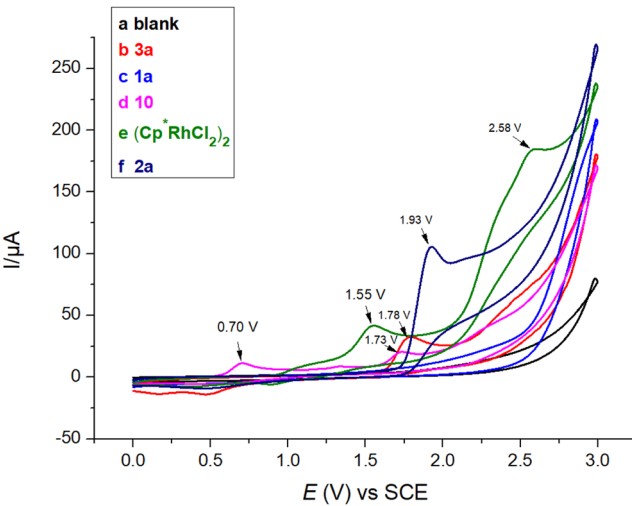

**Fig. 6 Cyclic voltametric study.** Cyclic voltammograms recorded on a Pt electrode (area = 0.03 cm²) with a scan rate of 100 mV s⁻¹: **a** MeCN containing 0.1 M n-Bu₄NPF₆; **b** MeCN containing 0.1 M n-Bu₄NPF₆, after addition of 4 mM **3a**; **c** MeCN containing 0.1 M n-Bu₄NPF₆, after addition of 4 mM **1a**; **d** MeCN containing 0.1 M n-Bu₄NPF₆, after addition of 4 mM **10**; **e** MeCN containing 0.1 M n-Bu₄NPF₆, after addition of 4 mM complex (Cp*RhCl₂)₂; **f** MeCN containing 0.1 M n-Bu₄NPF₆, after addition of 4 mM **2a**.

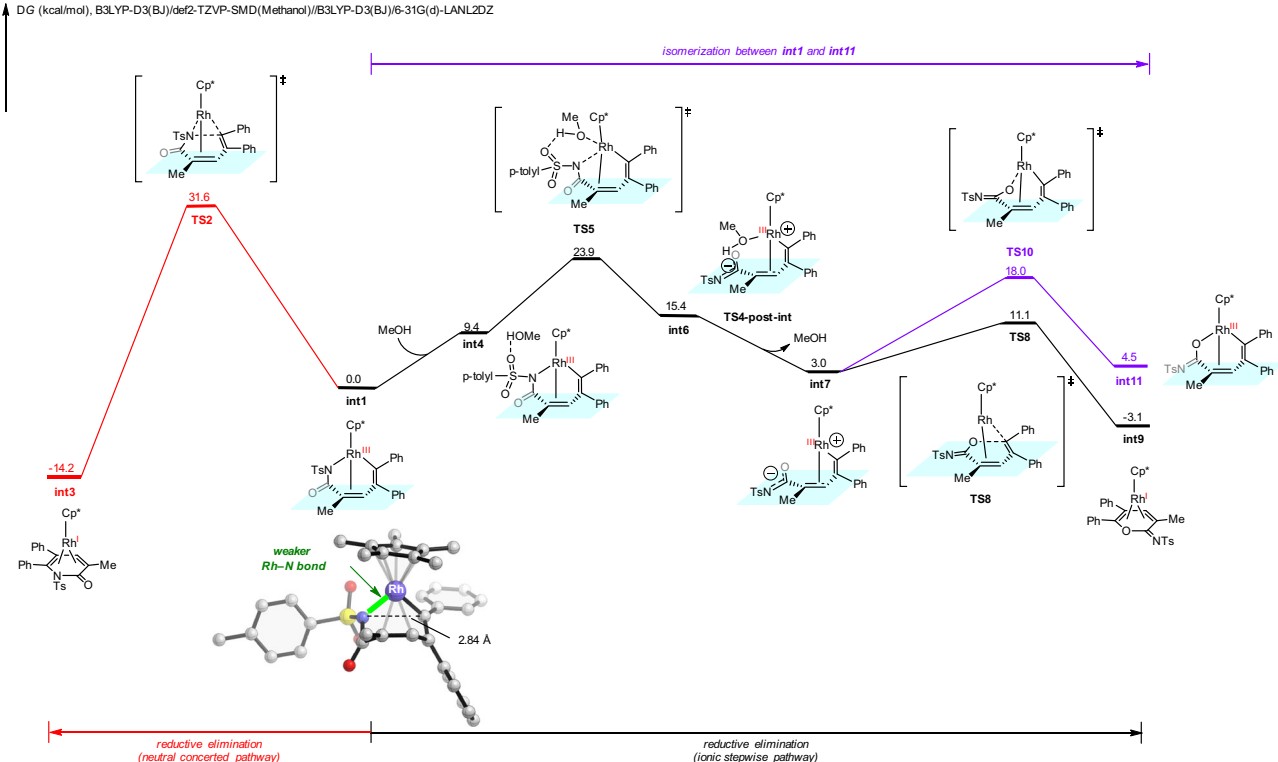

**Fig. 7 DFT calculations with 1a substrate.** DFT-computed free energy changes of competing reductive elimination pathways from seven-membered ring vinyl-rhodium intermediate when N-tosyl acrylamide **1a** was employed as substrate.

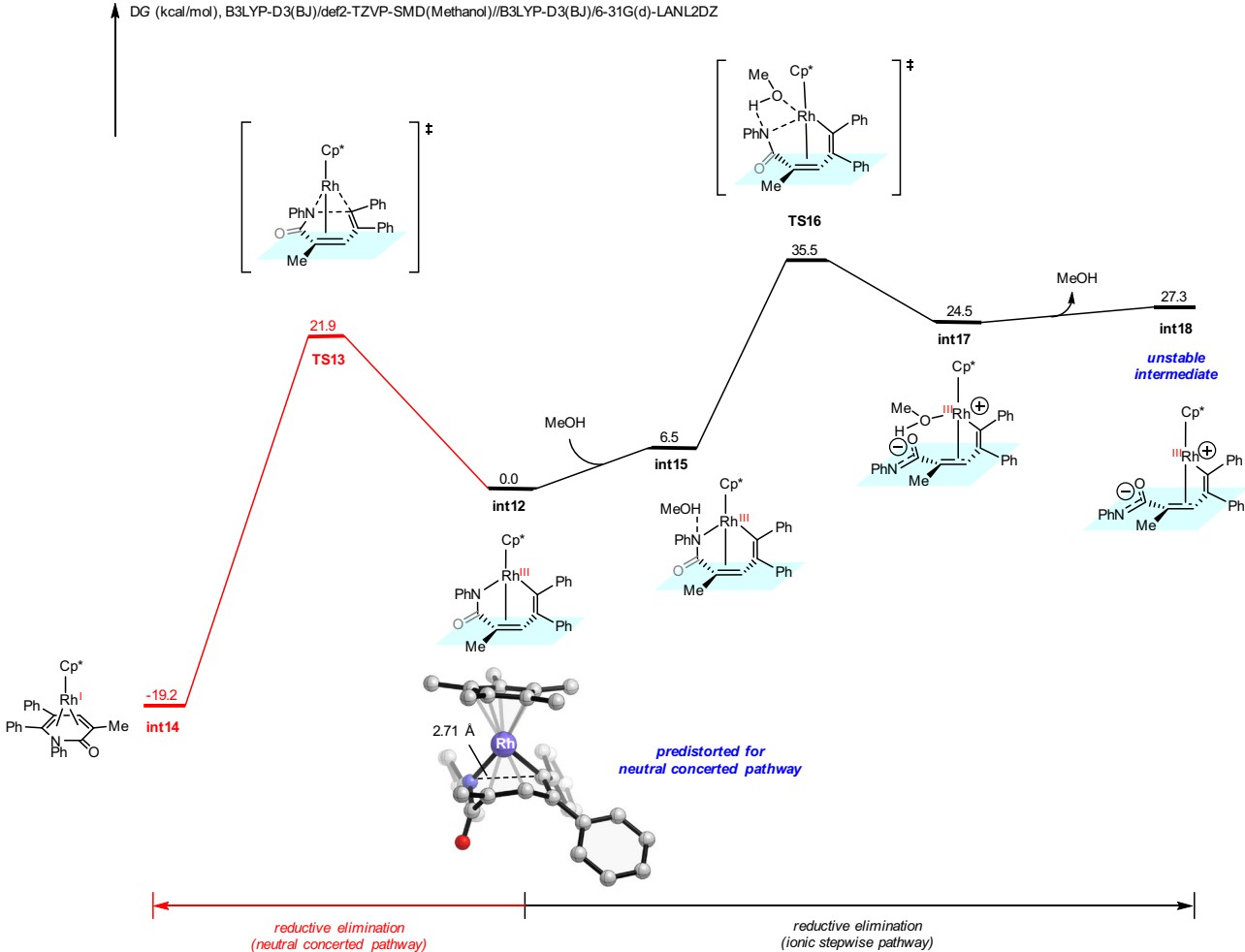

**Fig. 8 DFT calculations with 1b substrate.** DFT-computed free energy changes of competing reductive elimination pathways from seven-membered ring vinyl-rhodium intermediate **int10** when *N*-phenyl acrylamide **1b** was employed as substrate.

**Scope of α-pyridones**. We also examined the reactivity of a series of substituted acrylamides and alkynes for the synthesis of α-pyridones (Fig. 3). Acrylamides bearing a variety of functional groups such as alkyl, ester, ether, and aryl are well tolerated under the standard reaction conditions, affording α-pyridones in moderate to good yields (**4b**, **9a−9l**).

## Discussion

A series of experiments were carried out to elucidate the mechanism of this electrochemical C–H annulation reaction. First, acrylamide **1a** was subjected to the electrochemical C–H annulation reaction conditions in CH₃OD in the absence of an alkyne. Significant H/D exchange was observed, indicating that the putative C–H activation step is reversible (Fig. 4a). A kinetic isotope effect (KIE) value was determined by comparing parallel experiments using acrylamide **5e** and corresponding deuterated substrate **5e-*d₄*** (Fig. 4b). A KIE value of 1.4 was observed (see Supplementary information for details). In addition, we executed the stoichiometric reaction of acrylamides, diphenyla-cetylene **2a**, and (Cp*RhCl₂)₂ in the absence of electric current. To our delight, the rhodium sandwich complexes **10** and **11** were obtained in good yield, with the corresponding cyclic imidate as a neutral η4 ligand. Their structures were unambiguously confirmed by X-ray analysis (Fig. 5a). Upon anodic oxidation, the product **3a** is released from **10**, and is a coordinatively saturated, 18-electron complex (Fig. 5b). Additionally, **3a** is obtained in good yield when a catalytic amount of **10** is employed, which

suggests that **10** is a competent intermediate and catalyst in this electrochemical C–H annulation (Fig. 5c).

Complex **10** in 0.1 M solution of *n*-Bu₄NPF₆ in MeCN exhibits the first oxidation peak at 0.70 V versus saturated calomei elec-trode (curve d, Fig. 6), which is significantly lower than the oxidation potentials for the oxidation of other components in the reaction system (Fig. 6). This supports the hypothesis that the role of anodic oxidation is to oxidize a diene-Rh(I) complex to an active Rh(III) species with concomitant release of the product.

We next explored the reaction mechanism and the origins of substrate-controlled chemoselectivity through DFT calculations (see Supplementary information and Supplementary Data 1 for more details). From the active catalyst Cp*Rh(OAc)₂, sequential vinyl C–H activation of *N*-tosyl acrylamide **1a** and diphenylace-tylene insertion generate the seven-membered ring vinyl–rhodium intermediate **int1** (Supplementary Fig. 14 and Fig. 7)[79,80]. **Int1** can undergo competing reductive eliminations to form either the α-pyridone product or cyclic imidate (Fig. 7). The classic neutral concerted reductive elimination (red pathway) occurs through the three-membered ring transition state **TS2**, generating the α-pyridone product-coordinated complex **int3**. This neutral con-certed reductive elimination requires an insurmountable barrier of 31.6 kcal/mol, which is unfeasible under the experimental condi-tions. Alternatively, we found that the ionic stepwise pathway (black pathway) can be operative and produce the cyclic imidate product. This ionic stepwise pathway, discovered by Hong group[81] in similar transformation under ruthenium catalysis, initiates

**Fig. 9 Plausible catalytic cycle.** The catalytic cycle starts from C–H activation, and then alkyne insertion leads to the seven-membered ring vinyl-rhodium intermediate **D**. Subsequently intermediate **D** undergoes either the classic neutral concerted reductive elimination to produce intermediate **E**, or the ionic stepwise pathway to produce complex **10**. Finally, product formation under anodic oxidation, and complex **A** is regenerated.

through a heterolytic cleavage of the rhodium-nitrogen bond via **TS5** with the assistant of methanol to generate the zwitterionic intermediate **int6**. From **int6**, methanol dissociates to generate **int7**, subsequent facile C–O bond formation through **TS8** (IRC conformation of **TS8** is included in the Supplementary information) produces **int9**. The zwitterionic species **int7** also has the possibility for rhodium-oxygen bond formation via **TS10** (labeled in purple), but requiring a higher barrier as compared to the C–O bond formation. Comparing the free energy barriers of the two competing pathways, the ionic stepwise reductive elimination is more favorable by 7.7 kcal/mol (**TS2** vs. **TS5**), which is consistent with the experimental chemoselectivity favoring cyclic imidate when N-tosyl acrylamide is employed.

The DFT-computed free energy changes of the same competing reductive elimination pathways for the N-phenyl acrylamide **1b** substrate is shown in Fig. 8. From N-phenyl acrylamide **1b**, the sequential vinyl C–H activation and diphenylacetylene insertion generate the seven-membered ring vinyl-rhodium intermediate **int12** (Supplementary Fig. 15). This intermediate can undergo the classic neutral concerted reductive elimination pathway via **TS13** (red pathway), with a barrier of 21.9 kcal/mol. The alternative ionic stepwise pathway through **TS16** (IRC conformation of **TS16** is included in the Supplementary information) is significantly less favorable, due to the unstable zwitterionic species **int18**. Comparing to the tosyl substituent, the phenyl substituent significantly lowers the barrier of neutral concerted pathway while increases the barrier of ionic stepwise pathway, which results in the reversal chemoselectivity. For the ionic stepwise pathway, the electron-withdrawing tosyl substituent weakens the rhodium-nitrogen bond of **int1**, which favors its heterolytic cleavage and the generation of the zwitterionic intermediate **int7** (Fig. 7). The same process is endergonic by 27.3 kcal/mol for the N-phenyl substituted case (**int12** to **int18**, Fig. 8). This electronic effect is further supported by the computed rhodium-nitrogen bond

dissociation energies and additional Hammett analysis of the N-substitution (Supplementary Fig. 19). For the neutral concerted pathway, our distortion/interaction analysis revealed the distortion-controlled origins of the substituent effect (Supplementary Fig. 18). The phenyl substituent induces geometric change of the seven-membered rhodacycle in **int12**, leading to the predistortion towards the neutral concerted reductive elimination transition state **TS13**. This predistortion is reflected in the highlighted distance of the forming C–N bond in the seven-membered ring intermediates (2.71 Å in **int12**, Fig. 8; 2.84 Å in **int1**, Fig. 7). These insights provide a mechanistic basis for rational reaction designs in related transformations.

Based on our mechanistic studies, we propose a plausible catalytic cycle as shown in Fig. 9. Initially, C–H activation takes place to afford a cyclometallated Rh(III) intermediate **B**, following ligand exchange to deliver complex **C**. Next, migratory alkyne insertion results in the seven-membered rhodium complex **D**[82,83], which undergoes ionic stepwise or neutral concerted reductive elimination to give Rh(I) complex **10** or **E**. Intermediate **10** or **E** is a coordinately saturated, 18-electron complex. Upon anodic oxidation, the product is released from **10** or **E**, and complex **A** is regenerated.

In summary, we have developed an electrochemical method for the Rh(III)-catalyzed vinylic C–H annulation of acrylamides with alkynes. Owing to the robustness of this electrochemical C–H annulation, the reaction can be operated with IKA ElectraSyn 2.0 at room temperature, affording cyclic imidates with good to excellent yields. Additionally, divergent syntheses of a-pyridones and cyclic imidates are realized by simply switching the N-substitution of acrylamides. Furthermore, excellent regioselectivities are achieved with unsymmetrical alkynes, including terminal alkynes. Mechanistic and DFT studies combined to provide a rationale for the chemoselectivity switch and a basis for future reaction design in related transformations.

## Methods

**General procedure for the electrolysis**. The electrocatalysis was carried out in an IKA ElectraSyn 2.0 equipped with two platinum electrodes (each $0.8 \times 3.0$ cm$^2$). Acrylic amide (0.3 mmol, 1.5 equiv.), alkyne (0.2 mmol, 1.0 equiv.), $n$-Bu$_4$NOAc (0.6 mmol, 3.0 equiv.) and (Cp*RhCl$_2$)$_2$ (4.0 mol%, 99 wt%) were dissolved in MeOH (6.0 mL). Electrocatalysis was performed at room temperature with a constant current of 1.5 mA maintained for 7–12 h (2.0–3.4 F/mol). After the reaction, the mixture was concentrated in vacuo. The resulting residue was purified by silica gel flash chromatography to give the annulation product.

More experimental procedures and photographic guide for electrochemical C–H annulation are provided in the Supplementary information.

## Data availability

The X-ray crystallographic coordinates for structures reported in this article have been deposited at the Cambridge Crystallographic Data Centre (CCDC), under deposition number CCDC 1967777 (**7r**), CCDC 1967778 (**11**), CCDC 1967779 (**6c**), CCDC 1967780 (**10**), CCDC 1967781 (**3a**), CCDC 1967782 (**6o**), CCDC 1967783 (**7s**). The data can be obtained free of charge from The Cambridge Crystallographic Data Centre [http://www.ccdc.cam.ac.uk/data_request/cif]. The data supporting the findings of this study are available within the article and its Supplementary information files. Any further relevant data are available from the authors on request.

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

## Acknowledgements

This work was financially supported by the Strategic Priority Research Program of the Chinese Academy of Sciences (Grant XDB20000000), the NSF of China (Grants 21821002, 21772222, 91956112, 21702182, and 21873081), S&TCSM of Shanghai (Grants 17JC1401200 and 18JC1415600), the Fundamental Research Funds for the Central Universities (2020XZZX002-02), and the State Key Laboratory of Clean Energy Utilization (ZJUCEU2020007).

## Author contributions

Y.-K.X. and Q.-L.Y. discovered the reaction. X.-R.C. and S.-Q.Z. performed the DFT calculation. H.-M.G., X.H., and T.-S.M. directed the project. Y.-K.X., X.H., and T.-S.M. wrote the manuscript with input from all authors. All authors analyzed the results and commented on the manuscript.

## Competing interests

The authors declare no competing interests.
