## [Peer Review File. · Nature Communications]

REVIEWER COMMENTS

Reviewer #1 (Remarks to the Author):

Electrochemical synthesis has emerged as a comparatively environmentally benign alternative to conventional approaches to the oxidation or reduction of organic compounds because electric current is used instead of stoichiometric chemical oxidants or reductants. Thus, various metal-catalyzed electrochemical site-selective C–H functionalizations have been developed. However, electrochemical vinylic C–H functionalization is still rare. This fascinating manuscript from the Hong and Mei group represents the continuation of their recently developed program on site-selective C–H functionalization via the merger of electrochemistry and metal catalysis. Previous disclosure includes metal-catalyzed arene and alkane C–H functionalization via anodic oxidation and Ir-catalyzed vinylic C–H annulation of acrylic acids with terminal alkynes. In this manuscript, Hong and Mei group described Rh(III)-catalyzed electrochemical vinylic C–H annulation of acrylamides with alkynes in an undivided cell under mild reaction conditions. It is quite impressive that this electrochemical C–H annulation can be operated with IKA ElectraSyn 2.0 at room temperature, affording cyclic imidates with good to excellent yields. Furthermore, divergent syntheses of α -pyridones and cyclic imidates are realized by simply switching the N-substitution of acrylamides. Besides, excellent regioselectivities are achieved with unsymmetrical alkynes, including terminal alkynes. Finally, the authors carried out mechanistic and DFT studies to provide a rationale for the chemoselectivity switch. This discovery provides a basis for future reaction design in related transformations. The presentation of data in the manuscript is clear and concise. The Supporting Information is well organized and the experimental procedures and data are well written and clear. On the basis of these considerations, the impact of the work is worthy of publication in Nature Communications as an article after minor revision.

1. The authors propose the reoxidation of rhodium(I) to rhodium(III) being the rate-determining step. Do the authors have some evidence to support this hypothesis? In addition, the anode electrode could affect the reaction rate. I am curious whether other electrodes besides Pt work.
2. The reaction mostly tested standard α -substituted acrylamides in this study. Also, β -substituted (cinnamide-derived) acrylamides are common structures. Did the author have considered using β -substituted acrylamides as a coupling partner?
3. The consumed current (F mol⁻¹) is suggested to be included in the model reaction (Table 1).
4. How does the proposed mechanism rationalize the specific effect of the n-Bu₄NOAc? Do other anions affect the reaction?

Reviewer #2 (Remarks to the Author):

Suggestions for DFT results:

1. This outcome seems clear but some of the analysis based on electron effect is very simplistic. The authors should perform a detailed analysis of the electronic and steric effects, such as structure parameter, orbital, Fukui functions, *rdg* etc..
2. Either the energy axis in the profiles or the caption should include information about the phase and the solvent.
3. The methodology is fairly standard. However, I would seriously question why no dispersion correction has been included, since B3LYP suffers badly in this respect.

Reviewer #3 (Remarks to the Author):

In this submitted communication manuscript, Mei, Hong, and co-workers reported a sustainable electrochemical protocol for divergent Rh-catalyzed vinylic C–H annulation of acrylamides with alkynes. Using electrochemistry, the transformation happens under mild conditions without the necessity for stoichiometric metal oxidants and provides a synthetically useful method for the preparation of α -pyridones and cyclic imidates, with the later could be smoothly converted into α -pyrone derivatives with high yield. In addition, the substrate scope is much broader than previous

work that terminal alkynes are suitable and electron-deficient unsymmetric alkynes could afford the desired products with excellent regioselectivities. Notably, detailed mechanistic studies, especially DFT calculations, supported the proposed divergent reaction pathways caused by the N-substituent of the substrates. The preparation and investigation of the Rh sandwich complexes 10 and 11 indicated their competent roles in the catalytic cycles and the cyclic voltammetry studies supported the hypothesis that the anodic oxidation was employed to facilitate the regeneration of catalytically active Rh(III) species and the product liberation. Overall, this is an important contribution to the fields of electroorganic synthesis and transition metal-catalyzed C-H functionalization as a whole and revealed valuable mechanistic information for future reaction design. For the above reasons, I support the publication of this work in Nature Communications.

Below summarize some minor points need to be addressed before acceptance.

1) I am concerned about the use of Pt as anode for practical reasons that this material is not only very expensive but also might cause nanoparticle Pt contamination under electrolysis conditions, which is not good for large-scale synthesis and future adoption of this methodology in related fields.

2) The redox properties of substrate 1b are recommended to present in the paper. A simple CV scanning should be okay, and this information may shine some light on reaction mechanism with respect to the divergent selectivity.

Reviewer 1 suggested that “Publish in Nature Communications after minor revisions” and offered several suggestions.

- 1) The reviewer wanted us to provide some evidence to support that the reoxidation of rhodium(I) to rhodium(III) is the rate-determining step in our proposal mechanism.

Response: Thanks for this great suggestion. We further investigated the reaction mechanism, we examined the effect of current density on the reaction rate. Electrolysis was conducted respectively at different current at room temperature for different hours. (Page S111, Figure S10, Supporting Information). The current density has a great influence on the reaction rate, and the reaction rate increases as the current density increases. This result indicates that the reoxidation of rhodium(I) to rhodium(III) is the rate-determining step.

- 2) The reviewer asked whether other electrodes have been investigated.

Response: Thanks for this great suggestion. We have screened other electrodes such as graphite, BDD, glassy carbon and RVC (entries 11 and 12, Table 1, Page S35, Table S7, Supporting Information). These electrodes gave the slightly low yields (84–96% ¹H NMR yields, Page S35, Entries 2–4, Table S7, Supporting Information). We added a sentence which read as “Furthermore, changing the electrode material caused a small decrease in yield (entries 11 and 12).” in manuscript. And BDD as anode can react well (99% ¹H NMR yields). Similar results can be achieved by using graphite as cathode and anode at the same time (95% ¹H NMR yields).

- 3) The reviewer asked whether β-substituted acrylamides have been used as a coupling partner.

Response: Thanks for this suggestion. We did not observe any by-products for the reactions with these β-substituted acrylamides (Failed examples, Page S39, Supporting Information). The steric effects in β-substituted acrylamides could be the reason for observed lower reaction efficiency (**5s–5u**). In addition, we have included a sentence in the text to explain the observed low yields for β-substituted acrylamides, which read as “Unfortunately, β-substituted substrate like cinnamide-derived acrylamides give lower yields, which could be due to the steric effects (see SI for more details).”

- 4) The reviewer pointed out that “The consumed current (F mol⁻¹) is suggested to be included in the model reaction (Table 1).”

Response: Thanks for these suggestions. We have included consumed current in Table 1.

- 5) The reviewer wanted us to testify the influence of anion and provide some explanation for the critical role of acetate.

Response: Thanks for this great suggestion. We have screened other acetate salts or carboxylates, such as KOAc, NaOAc, KOPiv, NaOPiv·H₂O, (Page S31, Table S3, Supporting Information). These acetate salts gave the similar yields (86–94% ¹H NMR yields), but carboxylates gave slightly decreased yields (80–82% ¹H NMR yields). However, another bases like *n*-Bu₄NBF₄, *n*-Bu₄NCIO₄, *n*-Bu₄NPF₆, KPF₆, NaOTf is not effective for the reaction (7–28% ¹H NMR yield). These results revealed the crucial role for OAc⁻. The role of acetates in transition metal-catalyzed C–H functionalization has been well studied. In a C–H activation step, the deprotonation occurred through assistance of a coordinated acetate.

Reviewer 2 offered several suggestions for DFT results.

- 1) The reviewer pointed out that “This outcome seems clear but some of the analysis based on electron effect is very simplistic. The authors should perform a detailed analysis of the electronic and steric effects, such as structure parameter, orbital, Fukui functions, rdg etc.”

Response: We thank the reviewer for this advice. To further understand the origins of chemoselectivity, detailed analysis was performed during the revision.

For the ionic stepwise reductive elimination, the generation of the zwitterionic intermediate significantly favors the tosyl-substituted case. It requires 3.0 kcal/mol free energy to generate the tosyl-substituted zwitterionic intermediate **int7** from the seven-membered ring intermediate **int1**, while the same type of process requires 24.5 kcal/mol to generate the phenyl-substituted intermediate **int17** (Figure 1A). To verify the strength change of the rhodium-nitrogen bond, we computed the heterolytic bond dissociation energies on the model rhodium(III) complex **S22** (Figure 1B). The tosyl substitution stabilizes the dissociating amide anion, leading to a lower HBDE as expected (95.6 kcal/mol for tosyl vs. 108.5 kcal/mol for phenyl). To further corroborate the proposed electronic effect, the HBDEs of a series of substituted rhodium complexes were computed, and a linear relationship was identified with the corresponding Hammett constants (Figure 1C). These analyses provide additional supports for the proposed electronic origins of the *N*-substituent effect that electronic withdrawing substituent would promote the ionic stepwise reductive elimination pathway.

Figure 1. Analysis of the *N*-substituent effect on the ionic stepwise reductive elimination.

For the neutral concerted reductive elimination, the distortion/interaction analysis revealed the distortion-controlled origins of the *N*-substitution (Figure 2). The C–N reductive elimination transition states were separated to the rhodium catalyst part (labeled in red) and the amide substrate part (labeled in blue). The energy required to distort the selected fragment from the geometry in the pre-intermediate to the corresponding geometry in the transition state is the distortion energy, $\Delta E_{\text{dist-cat}}$ and $\Delta E_{\text{dist-sub}}$. The difference between the electronic energy barrier ΔE and the total distortion energy ΔE_{dist} is the stabilizing interaction energy between the two distorted fragments in the transition state, $\Delta E_{\text{int}} = \Delta E - (\Delta E_{\text{dist}} + \Delta E_{\text{dist-sub}})$.

The distortion/interaction analysis indicated that the distortion of the substrate fragment, $\Delta E_{\text{dist-sub}}$, is responsible for the substituent effect on the neutral concerted reductive elimination. The *N*-tosyl transition state **TS2** has a high $\Delta E_{\text{dist-sub}}$ of 21.3 kcal/mol (Figure 2A), while the $\Delta E_{\text{dist-sub}}$ of **TS13** is only 9.9 kcal/mol (Figure 2B). This suggests that the substrate fragment in the *N*-phenyl intermediate **int12** is predistorted towards the neutral concerted reductive elimination process. This predistortion is reflected in the highlighted C–N distances of **int1** (2.84 Å) and **int12** (2.71 Å). We believe that the electron-donating phenyl substituent enhances the rhodium-alkene d- π^* interaction in **int12**, which is supported by the rhodium-alkene distance change from **int1** (2.36 Å and 2.33 Å) to **int12** (2.28 Å and 2.27 Å). This induces the geometric change of the seven-membered ring in **int12** and results in the predistortion in favor of the neutral concerted C–N reductive elimination.

A. Distortion/interaction analysis of TS2

B. Distortion/interaction analysis of TS13

Figure 2. Analysis of the *N*-substituent effect on the neutral concerted reductive elimination.

- 2) The reviewer pointed out that “Either the energy axis in the profiles or the caption should include information about the phase and the solvent.”

Response: We have revised Figure 2 and Figure 3 to include the solvation information.

- 3) The reviewer pointed out that “The methodology is fairly standard. However, I would seriously question why no dispersion correction has been included, since B3LYP suffers badly in this respect.”

Response: All the DFT calculations have included the dispersion corrections using Grimme’s empirical approach (D3 version) with Becke-Johnson damping. We have also corrected the computational details to highlight this point in order to avoid misunderstandings.

Reviewer 3 suggested that “Publish in Nature Communications after addressed some minor points” and offered several suggestions:

- 1) From a more practical perspective, the reviewer asked whether other electrodes have been investigated.

Response: Thanks for this great suggestion. We have investigated other electrode materials such as graphite, BDD, glassy carbon and RVC (entries 11 and 12, Table 1, Page S35, Table S7, Supporting Information). The use of graphite or RVC as anode would cause slightly decreased in yield (91%, 84% ¹H NMR yields), and changed the anode to BDD or glassy carbon did not have significant influence on yield (99%, 96% ¹H NMR yields). In addition, the use of graphite as anode and cathode at the same time achieved similar result (95% ¹H NMR yields).

- 2) The reviewer wanted us to investigate the redox properties of substrate **1b** like CV.

Response: Thanks for this great suggestion. We have presented the CV of oxidation of *N*-phenyl acrylamide **1b** in Figure S13 (Page 114, Supporting Information).

REVIEWERS' COMMENTS

Reviewer #1 (Remarks to the Author):

The referee agreed to accept the revised manuscript.

Reviewer #2 (Remarks to the Author):

The manuscript has been improved greatly after revision and it is proper for publication.